# Hypotension after general anesthesia induction using remimazolam in geriatric patients: Protocol for a double-blind randomized controlled trial

**Masashi Yokose** [1]*, **Ryuki Takaki**[1], **Takahiro Mihara**[1,2], **Yusuke Saigusa**[3], **Natsuhiro Yamamoto**[1], **Kenichi Masui**[1], **Takahisa Goto**[1]

**1** Department of Anesthesiology and Critical Care Medicine, Yokohama City University Graduate School of Medicine, Yokohama, Japan, **2** Department of Health Data Science, Yokohama City University Graduate School of Data Science, Yokohama, Japan, **3** Department of Biostatistics, Yokohama City University Graduate School of Medicine, Yokohama, Japan

* yokose_p12@yahoo.co.jp

## Abstract

### Introduction

In geriatric patients, hypotension is often reported after general anesthesia induction using propofol. Remimazolam is a novel short-acting sedative. However, the incidence of hypotension after general anesthesia induction using remimazolam in geriatric patients remains unclear. This study aims to compare the incidence of hypotension associated with remimazolam and propofol in patients aged ≥80 years.

### Methods

This single-center, double-blind, randomized, two-arm parallel group, standard treatment-controlled, interventional clinical trial will include 90 patients aged ≥80 years undergoing elective surgery under general anesthesia who will be randomized to receive remimazolam or propofol for induction. The primary outcome is the incidence of hypotension after general anesthesia induction, occurring between the start of drug administration and 3 min after intubation. We define hypotension as mean blood pressure <65 mmHg. The primary outcome will be analyzed using the full analysis set. The incidence of hypotension in the two groups will be compared using the Mantel-Haenszel $\chi$2 test. Subgroup analysis of the primary outcome will be performed based on the Charlson comorbidity index, clinical frailty scale, hypertension in the ward, and age. Secondary outcomes will be analyzed using the Fisher's exact test, Student's *t* test, and Mann–Whitney *U* test, as appropriate. Logistic regression analysis will be performed to explore the factors associated with the incidence of hypotension after anesthesia induction.

### Discussion

Our trial will determine the efficacy of remimazolam in preventing hypotension and provide evidence on the usefulness of remimazolam for ensuring hemodynamic stability during general anesthesia induction in geriatric patients.

**Data Availability Statement:** All relevant data from this study will be made available upon study completion.

**Funding:** The authors received no specific funding for this work.

**Competing interests:** I have read the journal's policy and the authors of this manuscript have the following competing interests: Dr. Kenichi Masui was a consultant/advisor for Mundipharma K.K., Tokyo, Japan, and has been awarded a research grant for another study and has received payment for delivering domestic lectures from Mundipharma K.K. Dr. Takahisa Goto received honoraria from Mundipharma K.K. for chairing the lecture meeting. This does not alter our adherence to PLOS ONE policies on sharing data and materials.

## Trial registration

The study has been registered with UMIN Clinical Trials Registry (UMIN000042587), on June 30, 2021.

## Introduction

Remimazolam besylate is a novel short-acting drug that is rapidly metabolized by liver esterases [1, 2]. Its metabolites have a 410-fold lower affinity in humans [1]. Remimazolam has been approved for general anesthesia or procedural sedation [3].

Hypotension is a common side effect of general anesthesia induction, and its incidence in non-cardiac surgeries is reportedly 9–60% [4–7]. Decreased blood pressure, even for 5 min, can increase the risk of organ damage or morbidity [8–11]. In geriatric patients with declined physical and physiological function, careful anesthesia management is required to prevent hypotension after general anesthesia induction [4, 5]. Additionally, several procedures, such as mask ventilation, tracheal intubation, and drug administration, are performed, especially during the anesthesia induction. Therefore, anesthetic-induced hypotension should be avoided.

Propofol, the standard anesthetic for induction, has disadvantages such as vasodilation, reduced cardiac output [12, 13], and increased incidence of hypotension in high-risk patients aged >50 years [4]. Moreover, patients aged >80 years had a higher incidence of hypotension after anesthesia induction than patients aged <65 years [14]. Midazolam, a classical benzodiazepine, is used in patients with hemodynamic compromise to reduce the risk of hypotension after inducing anesthesia [15]. Therefore, remimazolam is expected to reduce the risk of hypotension during anesthesia induction in geriatric patients.

### Aim and objectives

We hypothesize that remimazolam, relative to propofol, will reduce the incidence of hypotension after anesthesia induction in geriatric patients. This study aims to examine the incidence of hypotension following anesthesia induction using remimazolam compared with propofol in patients aged ≥80 years. The study findings will aid drug selection for anesthesia induction in geriatric patients, regardless of their physical status.

## Material and methods

### Participants, interventions, and outcomes

**Trial design.** This is a single-center, double-blind, randomized, two-arm parallel group, standard treatment-controlled, interventional clinical trial. Block randomization of the participants, stratified by the presence/absence of hypertension with a 1:1 allocation will be performed.

Our protocol follows the Standard Protocol Items: Recommendations for Interventional Trials (SPIRIT) guidelines (S1 File) [16].

**Study setting.** This single center study will be conducted at the Yokohama City University Hospital, Japan.

**Eligibility criteria for participants.** The inclusion criteria are: (1) age ≥80 years with elective non-cardiovascular surgery planned under general anesthesia without regional anesthesia; (2) suitability of rapid induction of general anesthesia with oral tracheal intubation; (3)

absence of dementia and ability to provide informed consent; and (4) American Society of Anesthesiologists (ASA) physical status classes to .

The exclusion criteria are: (1) cerebral aneurysm or thoracic/abdominal aortic aneurysm; (2) predicted/history of difficulty in airway management; (3) allergies to drugs used in the study, i.e., remimazolam, propofol, and rocuronium; (4) undergoing maintenance dialysis; (5) severe liver dysfunction, i.e., Child–Pugh class C; (6) untreated/unstable ischemic heart disease; (7) severe aortic/mitral valvular disease; (8) arterial fibrillation, multiple atrial premature contraction, or ventricular premature contraction; (9) congestive heart failure (New York Heart Association classification  or ); (10) body mass index $\geq$30 kg.m$^{-2}$; (11) regular use of antipsychotics or antidepressant drugs; (12) mean blood pressure (MBP) <70 mmHg before general anesthesia induction; (13) failed first attempt of tracheal intubation; and (14) judged inappropriate for participation by the researchers.

**Interventions.** Standard monitors, such as a non-invasive blood pressure (NIBP) monitor, pulse oximeter, and electrocardiograph will be attached to patients. All vital signs will be measured with the IntelliVue™ MX800 and X2 monitors (Philips, Eindhoven, The Netherlands). The NIBP cuff (Soft Disposable Blood Pressure Cuff, Sun Tech Medical, Morrisville, NC, US) will be positioned on the arm contralateral to the arm with a peripheral intravenous (IV) catheter (a 20- or 22-gauge catheter inserted into the dorsal hand vein). A 10 cm extension tube (SA-ET211100, TERUMO, Tokyo, Japan) will be connected to the IV catheter, and a triple three-way stopcock (SA-3TR2, TERUMO, Tokyo, Japan) will be connected to the end of the extension tube. Ringer's acetate solution containing glucose 1% (Physio140, Otsuka Pharmaceutical, Tokushima, Japan) will be infused at a rate of 600 ml.hr$^{-1}$ using an infusion pump (TE-281A, TERUMO, Tokyo, Japan) until 3 min after intubation. The selected trial drug, remimazolam/propofol, will be administered to the IV line via a three-way stopcock closest to the patient. Before preoxygenation, the NIBP and heart rate were measured twice as baseline vital signs. Oxygen will be administered via face mask at a flow rate of 6 L.min$^{-1}$, and remifentanil will be initiated at 0.25 mcg.kg$^{-1}$.min$^{-1}$. After 3 min of preoxygenation, remimazolam (12 mg.kg$^{-1}$.h$^{-1}$ or propofol at 0.25 mg.kg$^{-1}$.10 s$^{-1}$ will be administered, until the patient loses consciousness, using a syringe pump (TERUFUSION syringe pump: TE-351, TERUMO, Tokyo, Japan). Concomitant use of anesthetics for inducing general anesthesia other than the trial drugs will be prohibited. After the patient loses consciousness, sevoflurane 1.5% will be initiated and rocuronium 0.6 mg.kg$^{-1}$) administered. Mask ventilation will be applied to maintain the end-tidal $CO_2$ at 35–45 mmHg. NIBP will be measured at 1-min intervals from the start of the anesthetic until 3 min after tracheal intubation. When hypotension, defined as MBP <65 mmHg, occurs, vasopressors will be administered as follows: a) ephedrine (4 mg) for hypotension with heart rate <80 beats.min$^{-1}$; b) phenylephrine (0.05 mg) hypotension with heart rate $\geq$80 beats.min$^{-1}$. To treat bradycardia (heart rate <45 beats.min$^{-1}$) without hypotension, atropine (0.5 mg) will be administered; to treat bradycardia with hypotension, ephedrine (4 mg) will be administered. Tracheal intubation using a McGRATH MAC video laryngoscope (Medtronic, Dublin, Ireland) will be performed by a blinded researcher (a board-certified anesthesiologist), 3 min after rocuronium administration. Successful tracheal intubation will be confirmed by waveform capnography. Immediately, the infusion rate of remifentanil will be reduced to 0.05 μg.kg$^{-1}$.h$^{-1}$. Except for securing tracheal intubation, contact with the patient will be prohibited until 3 min after intubation. Mechanical ventilation will be performed to maintain the end-tidal $CO_2$ at 35–45 mmHg.

The study protocol will be discontinued if the researchers judge difficulty in continuing the administration of trial drugs due to adverse events (AEs). No other criteria for dose reduction or withdrawal of trial drugs are set. Before recruitment, every unblinded researcher will be trained in the procedures such as drug preparation, administration, and the measurement of

NIBP and other vital signs. Anesthesia management will not be restricted, from 3 min after tracheal intubation.

**Outcomes.** The primary outcome is the incidence of hypotension (MBP <65 mmHg) from the start of anesthetic administration until 3 min after tracheal intubation.

The secondary outcomes are: (1) The maximum MBP, (2) minimum heart rate, and (3) number of vasopressor doses, administered from the start of anesthetic administration until 3 min after tracheal intubation. (4) Time from the start of anesthetic administration until loss of consciousness, defined as the loss of response to verbal stimuli (confirmed by calling the patient's name every 10 seconds from the start of anesthetic administration) and eyelash reflex. (5) The dose of anesthetic used for induction. (6) Incidence of pain after injection. Pain after the start of anesthetic administration will be inquired. Pain on injection will be estimated using the following four-point scale [17]: 0 = no pain, 1 = mild pain (only in response to questioning and without any behavioral signs), 2 = moderate pain (in response to questioning and accompanied by a behavioral sign, or pain reported spontaneously without questioning), and 3 = severe pain (strong vocal response and/or facial grimacing, arm withdrawal, or tears).

As safety outcome measures, AEs and serious adverse events (SAEs) based on medical records, examinations, and interviews will be analyzed.

**Participant timeline.** A blinded researcher will screen potential participants from the list of scheduled operations (with at least one day remaining before the surgery). All screened patients, including all excluded and enrolled patients, will be documented. From the recruited patients, written informed consent will be obtained by blinded researchers until the day before the surgery. Subsequently, NIBP will be measured for stratification. Allocation for the intervention will be performed by an unblinded researcher 30–60 min before each patient enters the operating room. All outcomes will be assessed and data collected according to participant timeline (Fig 1). Characteristics, such as age, sex, height, weight, medical history, and comorbidities; medicines, such as antihypertensives, sleep-inducing drugs, antipsychotics, and central nervous system agents without antidepressants; ASA physical status; clinical frailty scale [18]; surgical procedure; most recent laboratory values, including hemoglobin, creatinine, and serum albumin; and pulse rate will be recorded on the day before surgery. NIBP and heart rate in the operating room will be measured and collected at 1-minute intervals from the start of anesthetic administration until 3 min after tracheal intubation. AEs will be assessed the day after surgery.

**Sample size estimation.** Based on our institutional data, the frequency of hypotension after general anesthesia induction was estimated to be 40% in patients aged ≥80 years. The risk ratio of remimazolam to propofol for the frequency of hypotension associated with anesthesia induction was reported to be 0.28 [19]. If Fisher's exact probability test with a type I error set at 5% is performed, enrolling 84 patients could achieve a statistical power of 80%. Considering the probable dropout, inclusion of 90 patients is planned (n = 45 per group).

**Recruitment.** Multiple researchers will be readily available to obtain informed consent or for enrollment. To facilitate the cooperation of the anesthesiologists and other medical professionals involved in the surgeries in the facility, the research protocol will be thoroughly explained.

## Assignment of interventions

**Allocation.** A researcher (YS), not involved in research execution including statistical analysis, will generate the allocation sequence list with a 1:1 ratio of the treatment arms. The allocation sequence list will be according to computer-generated random numbers. Permuted-block randomization with stratification based on the presence/absence of hypertension,

|  | STUDY PERIOD | | | | | | |
|---|---|---|---|---|---|---|---|
|  | Enrollment | Allocation | Post-allocation | | | | Close-out |
| TIMEPOINT* | *1* | *2* | *3* | *4* | *5* | *6* | *7* |
| **ENROLLMENT:** | | | | | | | |
| *Eligibility screening* | X | | | | | | |
| *Informed consent* | X | | | | | | |
| *Allocation* | | X | | | | | |
| **INTERVENTIONS:** *[remimazolam]* | | | | X | | | |
| *[propofol]* | | | | X | | | |
| **ASSESSMENTS:** | | | | | | | |
| *Baseline variables [age, sex, height, weight, medical history, medicines, ASA-PS, CFS, surgical procedure, Laboratory values]* | X | X | | | | | |
| *Vital signs [blood pressure, heart rate]* | X | | X | X | X | X | |
| *Time until loss of consciousness* | | | | X | | | |
| *Injection pain* | | | | X | | | |
| *Dose of trial drug used for induction* | | | | | | X | |
| *Dose of vasopressors* | | | | | | X | |
| *AEs and SAEs* | | | | | | | X |

**Fig 1. Participant timeline.** *Timepoint 1: one day before surgery, Timepoint 2: after obtaining informed consent and till entering the operating room, Timepoint 3: in the operating room before anesthesia induction, Timepoint 4: from trial drug administration to loss of consciousness, Timepoint 5: from loss of consciousness to intubation, Timepoint 6: till 3 minutes after intubation, Timepoint 7: the day after the surgery. *Medicines* include antihypertensives, sleep inducing drugs, antipsychotics, and central nervous system agents without antidepressants. X indicates the timepoint at which each procedure in the protocol was carried out. *Laboratory values* include hemoglobin, creatinine, and albumin. *AE* adverse event, *ASA-PS* American society of Anesthesiology–physical status, *CFS* clinical frailty scale, *SAE* severe adverse event.

defined as systolic blood pressure (SBP) $\geq$140 mmHg or diastolic blood pressure $\geq$90 mmHg, in the ward, will be used. The presence/absence of hypertension will be determined based on the average of two NIBP measurements in the resting supine position using an automated blood pressure monitor (IntelliVue™ X2, Philips Health Care, Eindhoven, The Netherlands), in the ward. Three anesthesiologists (RT, NY, and YT), who are the unblinded researchers, will be responsible for allocation, drug preparation, and administration. The random allocation sequence list will be available only to the aforementioned four unblinded researchers; thus, the other researchers will be blinded to the allocation sequence and block size. After eligibility screening and acquiring informed consent by a blinded researcher, the registration center will reconfirm the eligibility and enroll the patients. On receiving the report of enrollment from the registration center, a blinded investigator will communicate with the unblinded researchers to allocate the patients.

**Blinding.** When obtaining written informed consent, patients will not be informed of the arm to which they will be assigned. Unblinded researchers will perform the allocation, drug preparation, and administration; they will neither be involved in other trial procedures such as screening, recruitment, data collection, or evaluating outcomes nor will they be attending

anesthetists. A syringe and a connected extension tube, filled with the traial drugs and mounted on a syringe pump, will be prepared by unblinded researchers. All of these components will be covered by an opaque plastic bag outside the operating room and brought into the operating room by an unblinded researcher. After laying the patient on the operating table, the entire IV line and puncture site on the patient's forearm will be covered with towels and opaque plastic sheets. The extension tube will be connected to a triple three-way stopcock in the IV line under the towels. The syringe pump will be operated from inside the opaque plastic bag. Subsequently, blinded researchers and attending anesthetists will enter the operating room and general anesthesia induction will be initiated. After data collection is completed in the operating room, the syringe pump and trial drug, still covered with an opaque plastic bag, will be returned to the location designated by the pharmacy department of our hospital. Patients, attending anesthetists, researchers evaluating outcomes, and other researchers will be blinded to the random allocation during the trial procedures. The random allocation sequence will not be exposed to the blinded researcher performing data analysis until its completion.

In the event of SAEs such as a medical emergency, which require identifying the intervention administered to the patient, blinded researchers are permitted to inquire with the unblinded researchers.

## Data collection, management, and analysis

**Data collection method.** All data collected during this study will be recorded in the respective patient case report form (CRF) by blinded researchers, trained in the procedure for collecting and evaluating outcomes before the start of the research. When a patient withdraws from the study, the reasons will be documented, and the dropout rate counted. Data from the dropout patients will also be collected up to the point of dropout.

**Data management.** Plausibility checks of CRF, to ensure the correctness and completeness of the data, will be performed by a blinded researcher. Inconsistencies in the data will be queried. Responses to queries will be documented directly in the CRF. By signing the CRF, the principal investigator will confirm that all procedures meet the protocol requirements, and complete and reliable data have been entered on the CRF. Two researchers will independently enter the data into the database. Consistency between the two entries will be tested. The database will be created using Microsoft Excel 2019 (Microsoft, Redmond, WA, US).

**Statistical methods.** The full analysis set will include all patients who will be enrolled, administered the trial/control drug, and whose outcome data will be obtained. The primary and secondary outcomes will be analyzed using the full analysis set. The safety analysis set will include all patients enrolled and administered the trial/control drug.

The incidence of hypotension, which is the primary outcome, in both the control and intervention groups will be calculated. The odds ratio (OR) with 95% confidence interval (CIs) between both groups will be calculated. The incidence of hypotension between the groups will be compared using the Mantel-Haenszel $\chi^2$ test and adjusted OR for the stratification factor for randomization. Homogeneity across strata will be assessed using the Breslow–Day test.

The following additional analyses of the primary outcome will be conducted: (a) Subgroups will be generated as follows and analyzed: 1) Charlson comorbidity index [20] $\leq 2$ or $\geq 3$; 2) clinical frailty scale $\leq 4$ or $\geq 5$; 3) presence or absence of hypertension (SBP $\geq 140$ mmHg or diastolic blood pressure $\geq 90$ mmHg) in the ward before the surgery; and 4) age $<90$ or $\geq 90$ years. (b) Sensitivity analysis of the primary outcome will be conducted: The OR with 95% CI adjusted for pre-induction blood pressure (SBP $\geq 160$ or $<160$ mmHg) will be calculated using the Mantel-Haenszel $\chi^2$ test. This cut-off point for SBP is 1) calculated based on the median value in the historical data of our institution for similar age groups as to the current protocol

and 2) determined with reference to the exclusion criteria for blood pressure in a previous study on the efficacy and safety of remimazolam [19]. (c) The absolute risk reduction values with 95% CI adjusted for the stratification factor for randomization will be calculated using Mantel-Haenszel risk differences.

Among the secondary outcomes, the incidence of injection pain will be considered as relative risk with 95% CIs and compared using Fisher's exact test. The maximum blood pressure after tracheal intubation, number of vasopressors used, dose of drugs for anesthesia induction, and time from anesthesia initiation until loss of consciousness will be given as median with inter-quartile ranges, and tested by Student's *t* test or Mann–Whitney *U* test, as appropriate. Logistic regression analysis, using body mass index, sex, age, ASA physical status, Charlson comorbidity index, clinical frailty scale, pre-induction hypertension, hypertension in the ward, use of antihypertensives and sleep-inducing drugs, serum albumin level, and types of anesthesia induction drugs, will be performed to explore the factors associated with hypotension incidence during anesthesia induction. For safety analysis, the incidence, type, and severity of AEs will be summarized for each group.

In all statistical analyses, the significance of the outcome will be defined as a two-tailed P-value <0.05. Patients with missing values for each outcome will be excluded from the final analysis. All statistical analyses will be performed using the R software (R Foundation for Statistical Computing) and Prism 7.0 (GraphPad, Inc., San Diego, CA, US). Interim analyses have not been planned.

## Monitoring

**Data monitoring and auditing.**   The decision of our institutional ethics board, based on the procedures in the institutional manual, was not to conduct auditing and monitoring. However, the board requested that self-inspection be conducted every 3 months as an alternative to data monitoring.

**Safety considerations.**   Safety assessments will be composed of monitoring vital signs during anesthesia and observing and recording all AEs and SAEs. AEs are defined as all unfavorable/unintended medical events that occur in patients, whether causally related to the research protocol or not. For details on evaluating or reporting AEs and SAEs, see S2 File.

All possible side effects of remimazolam and propofol are described in the summary on the product package inserts. The incidence of adverse reactions to remimazolam is reportedly similar to that of propofol [19]. Any additional harm associated with study participation, other than the common AEs in routine clinical procedure, are not expected in either group. If such an event occurs due to trial participation, researchers will provide appropriate medical treatment.

## Ethics and dissemination

**Ethics approval and consent to participate.**   The study protocol (S3 and S4 Files) was approved by the Institutional Ethics Committee of Yokohama City University Hospital, Yokohama, Japan on February 15, 2021 (approval number: B210204001). Written informed consent will be obtained before enrollment from all patients. Voluntary willingness of all participants will be ascertained after providing comprehensive written and verbal explanations regarding the study.

**Protocol amendments.**   If the research protocol needs to be revised, the principal investigator will discuss with other researchers. The principal investigator will submit the revised protocol to the ethics board and chairperson of the research institution. After review and approval by the institutional ethics board, permission from the chairperson of the research institution will be obtained.

**Confidentiality.** All patients will be identified by individual registry numbers, to anonymize CRFs and other documents, at the time of enrollment. This individual registry number consists of numbers/letters unrelated to the information that can identify a specific individual. The principal investigator creates and maintains the list that links the anonymized data to each patient. Under the supervision of the principal investigator, all data and documents will be secured from unauthorized access in locked cabinets with restricted access. Data analysis will be performed using anonymized data.

**Access to data.** After the data collection is completed, selected researchers will be granted full access to the database.

**Dissemination policy.** The study results will be presented at appropriate international scientific journals and scientific conferences. A professional writing service will not be engaged.

**Status and timeline of the study.** This study has been recruiting since June 30, 2021. It is estimated that recruitment will end in July 2022. The protocol in use is version 1.1, dated April 30, 2021.

## Discussion

Hypotension after general anesthesia induction is often encountered by anesthesiologists in routine clinical practice. Various reports demonstrate increased age as risk factors for hypotension [4–6]. MBP <80 mmHg sustained >10 min or MBP <70 mmHg for a shorter duration mildly increased the risk of end-organ damage [10]. An assessed MBP <55 mmHg predicted adverse cardiac outcomes and acute kidney injury [9]. Intraoperative hypotension with MBP <55 mmHg in geriatric patients is a risk factor for delirium [21], associated with declined activities of daily living and postoperative mortality [22]. Midazolam, a benzodiazepine, is less likely than propofol to cause significant hypotension [23–25]; consequently, remimazolam, a novel benzodiazepine, is less likely to cause hypotension. Midazolam is not an ideal anesthetic induction agent, especially for short duration surgeries in geriatric patients, owing to its high individual variability [26, 27] and the residual effect of midazolam and its primary metabolite, 1-hydroxymidazolam [28, 29]. Remimazolam is expected to have less pharmacodynamic inter-individual variability than midazolam because of its short action [30] and the 410-fold lower affinity of its primary metabolite, CNS-7054, than remimazolam [1]. Therefore, we hypothesized that it would be a more suitable for anesthesia induction than others in geriatric patients.

Various definitions for intraoperative hypotension have been previously reported; however, an established standard definition is lacking [31]. It has been suggested that even brief durations of intraoperative MBP <65 mmHg are associated with organ damage and perioperative mortality [10, 11]; hence, and maintaining MBP above 60–70 mmHg may reduce these risks [11]. Although the appropriate target MBP for geriatric patients is also not clearly defined, we believe that our definition of hypotension (MBP $\geq$65 mmHg), is an acceptable threshold for safety.

Our protocol adopts an infusion rate of 12 mg.kg$^{-1}$.hr$^{-1}$ for remimazolam and 0.025 mL.kg$^{-1}$.10 sec$^{-1}$ for propofol, because the institutional ethics board did not approve deviation from the description in the package insert. The optimal infusion rate of remimazolam for general anesthesia induction in geriatric patients has not been clarified. The product package insert for remimazolam states that a decreasing infusion rate should be considered for administration to geriatric patients; however, the infusion rate was not specified. Previously, no statistically significant difference in the incidence of decreased blood pressure in high-risk surgical patients between remimazolam infusion rates of 6 mg.kg$^{-1}$.hr$^{-1}$ and 12 mg.kg$^{-1}$.hr$^{-1}$ were reported [32]. Contrarily, marked sensitivity to propofol was reported in geriatric patients [33, 34]. Therefore, during propofol administration to geriatric patients, reducing the infusion rate to half, e.g., 0.025 mL.kg$^{-1}$.10 sec$^{-1}$, should be considered.

The strength of our study is the focus on older patients within the geriatric population. In 2021, 29.1% of the total Japanese population were aged ≥65 years and 9.6% were aged ≥80 years, one of the highest among all countries [35]. It is soon expected that general anesthesia for older patients within the geriatric population would be a routine requirement. Our study could support clinical decision making regarding the choice of safer drugs, in patients at a higher risk of hypotension, during general anesthesia induction.

## Limitations of the study design

This study has several limitations. First, it will not include high-risk patients who may require more careful management for hemodynamic stabilization (described in the exclusion criteria) owing to ethical concerns. Second, our study will not estimate long-term patient outcomes, such as mortality or morbidity. Finally, it is difficult to determine whether the doses of propofol and remimazolam selected for general anesthesia induction are equivalent because the hypnotic potency ratio of propofol to remimazolam in patients aged ≥80 years is still unclear. Our study protocol adopted an alternative approach, in which administration will be terminated at the time of consciousness loss. However, because this method only regulates the rate of administration, the dose of anesthetic administered in this study may be higher than that used in routine clinical practice.

Our single-center, prospective, double-blind randomized control trial will reveal the efficacy of remimazolam in preventing hypotension. This would provide scientific evidence supporting the use of remimazolam as a method to contribute to hemodynamic stability during general anesthesia induction in geriatric patients.

## Supporting information

**S1 File. SPIRIT checklist.**
(DOC)

**S2 File. Definition and evaluation of adverse events.**
(DOCX)

**S3 File. Study protocol in English.**
(DOCX)

**S4 File. Study protocol in Japanese.**
(DOCX)

## Acknowledgments

We thank Yusuke Mizuno, Tomoya Irie, Yusuke Nagamine, Kenta Okamura, Kentaro Tojo, Masaru Kikuchi, Hiroyuki Tanaka, Shunsuke Takaki, and Yusaku Terada from the Department of Anesthesiology and Critical Care Medicine, Yokohama City University Hospital, Yokohama, Japan, and Misako Tomata and Tomoya Miura from Yokohama City University Hospital, Yokohama, Japan, for their cooperation while developing the protocol.

## Author Contributions

**Conceptualization:** Masashi Yokose, Ryuki Takaki, Takahiro Mihara, Yusuke Saigusa, Natsuhiro Yamamoto, Kenichi Masui, Takahisa Goto.

**Supervision:** Takahisa Goto.

**Writing – original draft:** Masashi Yokose.

**Writing – review & editing:** Masashi Yokose, Ryuki Takaki, Takahiro Mihara, Yusuke Saigusa, Natsuhiro Yamamoto, Kenichi Masui, Takahisa Goto.

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
