## [Decision Letter · Decision Letter 0]

20 Jul 2022

PONE-D-22-13552Hypotension after general anesthesia induction using remimazolam in geriatric patients: Protocol for a double-blind randomized controlled trialPLOS ONE

Dear Dr. Yokose,

Thank you for submitting your manuscript to PLOS ONE. After careful consideration, we feel that it has merit but does not fully meet PLOS ONE’s publication criteria as it currently stands. Therefore, we invite you to submit a revised version of the manuscript that addresses the points raised during the review process.

We look forward to receiving your revised manuscript.

Kind regards,

Jamie Royle

Staff Editor

PLOS ONE

Journal Requirements:

Reviewers' comments:

Reviewer's Responses to Questions

**Comments to the Author**

1. Does the manuscript provide a valid rationale for the proposed study, with clearly identified and justified research questions?

Reviewer #1: Yes

2. Is the protocol technically sound and planned in a manner that will lead to a meaningful outcome and allow testing the stated hypotheses?

Reviewer #1: Yes

3. Is the methodology feasible and described in sufficient detail to allow the work to be replicable?

Reviewer #1: Yes

4. Have the authors described where all data underlying the findings will be made available when the study is complete?

Reviewer #1: Yes

5. Is the manuscript presented in an intelligible fashion and written in standard English?

Reviewer #1: Yes

6. Review Comments to the Author

You may also provide optional suggestions and comments to authors that they might find helpful in planning their study.

Reviewer #1: Dear colleagues

Thank you for submitting the manuscript entitled: Hypotension after general anesthesia induction using remimazolam in geriatric patients: Protocol for a double-blind randomized controlled trial to PLOS ONE journal. I read this amazing manuscript and I think you should consider the following points about your manuscript.

1. In the method you wrote, patients aged more than 80 years therefore is it possible these patients were pregnant? As you wrote in the exclusion criteria?

2. How could you blind this study because the colors of propofol and remimazolam are different, please explain more about this.

Best regards.

7. PLOS authors have the option to publish the peer review history of their article (what does this mean?). If published, this will include your full peer review and any attached files.

Reviewer #1: No

---

## [Author Response · Author response to Decision Letter 0]

13 Aug 2022

Dear Editor,

We greatly appreciate the editor and reviewers’ comments regarding our manuscript, titled "Hypotension after general anesthesia induction using remimazolam in geriatric patients: Protocol for a double-blind randomized controlled trial” (PONE-D-22-13552). We have revised our manuscript according to the comments by the reviewers. Our responses to the comments are described below in a point-by-point manner.

Reviewer #1:

Reviewer’s comment:

1. In the method you wrote, patients aged more than 80 years therefore is it possible these patients were pregnant? As you wrote in the exclusion criteria?

Our response:

We agree with the reviewer’s comment that the exclusion criterion was not correctly phrased. We have removed the term “pregnancy” from the exclusion criterion, and renumbered the list as follows (Lines 109-119 of the revised manuscript). 

“The exclusion criteria are: (1) cerebral aneurysm or thoracic/abdominal aortic aneurysm; (2) predicted/history of difficulty in airway management; (3) allergies to drugs used in the study, i.e., remimazolam, propofol, and rocuronium; (4) undergoing maintenance dialysis; (5) severe liver dysfunction, i.e., Child–Pugh class C; (6) untreated/unstable ischemic heart disease; (7) severe aortic/mitral valvular disease; (8) arterial fibrillation, multiple atrial premature contraction, or ventricular premature contraction; (9) congestive heart failure (New York Heart Association classification Ⅲ or Ⅳ); (10) body mass index ≥30 kg.m-2; (11) regular use of antipsychotics or antidepressant drugs; (12) mean blood pressure (MBP) <70 mmHg before general anesthesia induction; (13) failed first attempt of tracheal intubation; and (14) judged inappropriate for participation by the researchers.”

Reviewer’s comment:

2. How could you blind this study because the colors of propofol and remimazolam are different, please explain more about this. 

We thank the reviewer for this constructive comment that will help us clarify our research protocol further. We have added a detailed description of the blinding process as follows (Lines 249-258 of the revised manuscript).

“A syringe and a connected extension tube, filled with trial drugs and mounted on a syringe pump, will be prepared by unblinded researchers. All of these components will be covered by an opaque plastic bag outside the operating room and brought into the operating room by an unblinded researcher. After laying the patient on the operating table, the entire IV line and puncture site on the patient's forearm will be covered with towels and opaque plastic sheets. The extension tube will be connected to a triple three-way stopcock in the IV line under the towels. The syringe pump will be operated from inside the opaque plastic bag. Subsequently, blinded researchers and attending anesthetists will enter the operating room and general anesthesia induction will be initiated.”

---

## [Decision Letter · Decision Letter 1]

19 Sep 2022

Hypotension after general anesthesia induction using remimazolam in geriatric patients: Protocol for a double-blind randomized controlled trial

PONE-D-22-13552R1

Dear Dr. Yokose,

We’re pleased to inform you that your manuscript has been judged scientifically suitable for publication and will be formally accepted for publication once it meets all outstanding technical requirements.

Kind regards,

Walid Kamal Abdelbasset, Ph.D.

Academic Editor

PLOS ONE

Additional Editor Comments (optional):

Reviewers' comments:

Reviewer's Responses to Questions

**Comments to the Author**

1. Does the manuscript provide a valid rationale for the proposed study, with clearly identified and justified research questions?

Reviewer #1: Yes

2. Is the protocol technically sound and planned in a manner that will lead to a meaningful outcome and allow testing the stated hypotheses?

Reviewer #1: Yes

3. Is the methodology feasible and described in sufficient detail to allow the work to be replicable?

Reviewer #1: Yes

4. Have the authors described where all data underlying the findings will be made available when the study is complete?

Reviewer #1: Yes

5. Is the manuscript presented in an intelligible fashion and written in standard English?

Reviewer #1: Yes

6. Review Comments to the Author

You may also provide optional suggestions and comments to authors that they might find helpful in planning their study.

Reviewer #1: Dear colleagues

Thank you for submitting the revised manuscript entitled: Hypotension after general anesthesia induction using remimazolam in geriatric patients: Protocol for a double-blind randomized controlled trial PLOS ONE journal. I read the revision of the manuscript, and I think right now it is more acceptable.

Best regards

7. PLOS authors have the option to publish the peer review history of their article (what does this mean?). If published, this will include your full peer review and any attached files.

Reviewer #1: No

---

## [Editor Report · Acceptance letter]

21 Sep 2022

PONE-D-22-13552R1 

Hypotension after general anesthesia induction using remimazolam in geriatric patients: Protocol for a double-blind randomized controlled trial 

Dear Dr. Yokose:

I'm pleased to inform you that your manuscript has been deemed suitable for publication in PLOS ONE. Congratulations! Your manuscript is now with our production department. 

Kind regards, 

on behalf of

Dr. Walid Kamal Abdelbasset 

Academic Editor

PLOS ONE